# Room-temperature quantum nanoplasmonic coherent perfect absorption

Yiming Lai[1,8], Daniel D. A. Clarke [1,8], Philipp Grimm [2], Asha Devi[1], Daniel Wigger[1], Tobias Helbig [3], Tobias Hofmann [3], Ronny Thomale [3], Jer-Shing Huang [4,5,6,7], Bert Hecht [2] ✉ & Ortwin Hess [1] ✉

Light-matter superposition states obtained via strong coupling play a decisive role in quantum information processing, but the deleterious effects of material dissipation and environment-induced decoherence inevitably destroy coherent light-matter polaritons over time. Here, we propose the use of coherent perfect absorption under near-field driving to prepare and protect the polaritonic states of a single quantum emitter interacting with a plasmonic nanocavity at room temperature. Our scheme of quantum nanoplasmonic coherent perfect absorption leverages an inherent frequency specificity to selectively initialize the coupled system in a chosen plasmon-emitter dressed state, while the coherent, unidirectional and non-perturbing near-field energy transfer from a proximal plasmonic waveguide can in principle render the dressed state robust against dynamic dissipation under ambient conditions. Our study establishes a previously unexplored paradigm for quantum state preparation and coherence preservation in plasmonic cavity quantum electrodynamics, offering compelling prospects for elevating quantum nanophotonic technologies to ambient temperatures.

Quantum electrodynamic strong coupling lies at the heart of contemporary research efforts in condensed-matter quantum optics, representing both a fundamentally profound and technologically compelling regime of light-matter interaction[1–3]. In the strong coupling regime, the confined photonic mode of an optical resonator becomes inextricably intertwined with the electronic or phononic excitations of matter, giving rise to *dressed* or *polariton* states that exhibit a dual light-matter character[4–7]. Polaritons form whenever the timescale for energy exchange between the light and matter components becomes shorter than the intrinsic decay or decoherence times; they manifest spectrally via a so-called vacuum Rabi splitting in the scattering or photoluminescence emission from the coupled system, while their characteristic temporal energy cycling is revealed via time-resolved detection of Rabi oscillations[5,8]. Access to the strong coupling regime in the single-emitter limit has emerged as a particularly promising resource for nascent photonic quantum information processing strategies, supporting quintessential functionalities like single-qubit coherent control[9,10], ultrafast single-photon emission[11] and optical switching[12], as well as quantum sensing[13]. In tandem, the strong coupling between optical modes and electronic or phononic transitions in the collective many-emitter regime has led to the observation of striking modifications in exciton transport[14], polaron photoconductivity[15] and ground-state chemical reactivity[16], paving the way towards the manipulation of non-equilibrium material properties

[1]School of Physics and CRANN Institute, Trinity College Dublin, Dublin 2, Ireland. [2]Nano-Optics & Biophotonics Group, Department of Experimental Physics 5, and Röntgen Research Center for Complex Material Research, Physics Institute, University of Würzburg, Am Hubland, Würzburg, Germany. [3]Theoretische Physik I, Julius-Maximilians-Universität Würzburg, Am Hubland, Würzburg, Germany. [4]Leibniz Institute of Photonic Technology, Albert-Einstein Strasse 9, Jena, Germany. [5]Institute of Physical Chemistry and Abbe Center of Photonics, Friedrich-Schiller-Universität Jena, Helmholtzweg 4,, Jena, Germany. [6]Research Center for Applied Sciences, Academia Sinica, 128 Sec. 2, Academia Road, Taipei, Nankang, Taiwan. [7]Department of Electrophysics, National Chiao Tung University, Hsinchu, Taiwan. [8]These authors contributed equally: Yiming Lai, Daniel D. A. Clarke. ✉e-mail: hecht@physik.uni-wuerzburg.de; ortwin.hess@tcd.ie

based on the delocalized character of the mixed light-matter states and thus promoting a new generation of cavity-enhanced optoelectronic devices and polariton-enabled chemistry protocols[17,18].

Harnessing the potential of strong-coupling-enabled photonic quantum technologies poses a twofold challenge, namely to selectively prepare and to temporally sustain or protect the hybrid light-matter states. In principle, accessing and preserving the coherent character of specific polaritons for arbitrary temporal intervals could facilitate unprecedented single-emitter manipulation for quantum information processing tasks[19], enabling precise state initialization for quantum logic operations[20], extending quantum information storage times via cavity-emitter entanglement[21], or allowing one to harness the coherent multiphoton dynamics of the anharmonic Jaynes-Cummings ladder for non-linear quantum optical functionalities[22,23]. Yet, in spite of its manifest importance, few studies have revealed plausible strategies for selectively exciting and temporally sustaining polaritonic states. In the context of semiconductor quantum optics[24,25], strongly coupled states of quantum dot excitons and microcavity photons are generally excited in a non-selective manner with broadband laser light or incoherent pumping, and temporally sustained through the use of high-quality-factor (Q) resonators. Although, for example, photonic crystal defect cavities with unprecedented Q factors (exceeding $11 \times 10^6$) have been fabricated[26], the resulting photon storage times typically remain on the order of nanoseconds. More fundamentally, all dielectric microcavity designs suffer a common limitation in that their supported mode volumes $V_m$ are unavoidably bounded by the diffraction limit, which in turn constrains the attainable single-emitter coupling strengths via $g \propto 1/\sqrt{V_m}$[5].

In stark contrast, plasmonic nanoresonators offer the unique ability to confine light to extremely sub-wavelength volumes and massively enhance local electromagnetic fields via resonant surface plasmon modes, thereby constituting exceptional architectures for enhanced light-matter interaction and the exploration of extreme nano-optics[27]. In particular, room-temperature strong coupling using single molecules and colloidal quantum dots in nanoplasmonic environments has been realized using ultrathin (~ 1-nm), metal-insulator-metal nanocavities[13,28–30] and scanning probe tips[31,32], whose ultralow mode volumes of $V_m < 100$ nm$^3$ promote high coupling strengths $g$ and thereby effectively overcome the adversarial effects of intrinsic Ohmic dissipation, dephasing and radiative loss. However, the plasmon-exciton polariton lifetime is generally limited by the ultrafast decay of the cavity plasmon mode, typically on a timescale of 10–100 fs. Although such ultrafast plasmonic near-field evolution can be exploited to achieve high-speed quantum operations, including

dynamic bi- and tripartite entanglement in quantum dots[33,34], it is nevertheless imperative to explore pathways for improving the temporal robustness of strongly coupled plasmon-emitter states under ambient conditions, with the aim of realizing truly room-temperature-viable quantum nanophotonic devices.

In this article, we propose and theoretically explore a novel strategy for selective preparation and even potential "immortalization" of selected plasmon-exciton polariton states by means of quantum nanoplasmonic coherent perfect absorption. Coherent perfect absorption (CPA)[35–40] is the time-reversed analog of lasing at the threshold and is thus an equally fascinating process as lasing itself. In order to achieve lasing, the gain supported by the active medium within an optical resonator must be sufficiently high to reach the lasing threshold and thereby give rise to a coherent outgoing mode. In contrast, for a CPA system, the intrinsic losses of an optical resonator must possess a certain critical value to achieve the total reflectionless absorption of an incoming coherent mode. Although originally conceptualized and explored in the domain of classical optics, more recent work has sought to elucidate and harness CPA in the quantum regime. For example, modulation between total absorption and transmission in deeply subwavelength films at the single-photon level[41,42], single- and two-photon absorption in two-photon N00N and polarization-entangled states[43–45], as well as the implementation of an anti-Hong-Ou-Mandel effect[46] have been explored, while CPA-based single-photon detectors[47,48], cavity-free quantum-optical memories[49] and entanglement generation in quantum networks[50] promote CPA as a candidate for innovative quantum photonic technologies.

Previously, it has been established that for plasmonic nanoresonators, which exhibit both Ohmic and radiative losses, a generalized formulation of classical CPA (gCPA) can be invoked for a rigorous analysis of near-field energy absorption and perfect impedance matching in plasmonic nanotechnology. In particular, it has been suggested that gCPA can prevent a plasmonic mode from decaying, by coherently compensating the sum of all losses[51]. In an active system, where the plasmonic resonator couples to a quantum emitter, we might envisage the quantum analog of gCPA, namely quantum nanoplasmonic coherent perfect absorption (qnCPA), whose potential for hybrid light-matter quantum technologies has yet to be fully elucidated. In our present work, we explore a unique principle for selectively initializing and preserving the coherence of strongly coupled light-matter states by leveraging the frequency-specific, unidirectional and non-perturbing energy transfer characteristics of qnCPA. Adopting the system architecture illustrated in Fig. 1, we demonstrate that under plasmonic single-mode waveguide driving, the qnCPA regime

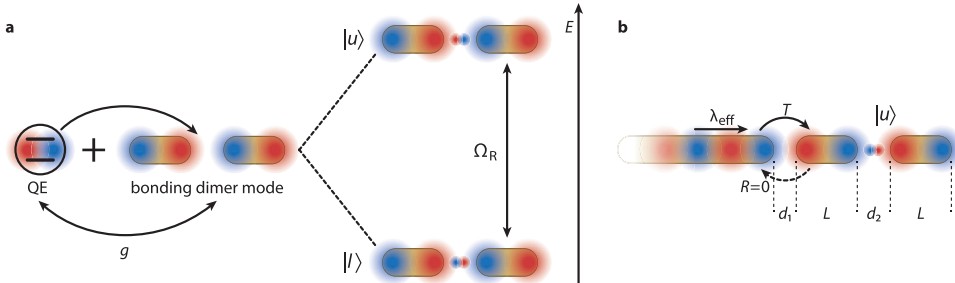

**Fig. 1 | Preparing strongly coupled light-matter states via quantum nanoplasmonic coherent perfect absorption (qnCPA). a** Schematic diagram of a dimer nanocavity comprising a pair of end-to-end aligned gold nanorods (diameter 30-nm) with a quantum emitter (QE) positioned at the center of the gap. The blue and red areas sketch the amplitudes and phases of the resonator bonding plasmon mode and QE near-fields. Strong coupling of the two-level system and the nanocavity plasmon (coupling strength $g$) gives rise to a pair of upper ($|u\rangle$) and lower ($|l\rangle$) polariton states with vacuum Rabi splitting $\Omega_R$. Note the difference in relative oscillation phase between the QE and nanocavity fields for the lower and upper

polariton states. **b** Illustration of the considered, waveguide-driven nanocavity-QE system for qnCPA, consisting of a cylindrical, semi-infinite gold nanowire (diameter 30-nm) with a hemispherical termination coupled to the nanocavity-QE device. By fine tuning the nanorod lengths $L$ and gap sizes $d_1$, $d_2$, the individual polariton states (here illustrated for $|u\rangle$) can be selectively initialized via coherent perfect absorption of the plasmonic nanowire mode at effective wavelength $\lambda_{eff}$, characterized by zero reflectance $R$ and maximal transmittance $T$. In principle, by harnessing the coherent, unidirectional and non-perturbing near-field energy transfer in a regime of exact loss compensation, the strongly coupled state could also be preserved in time.

can lock a nanocavity-emitter system in either the upper, $|u\rangle$, or lower, $|l\rangle$, plasmon-emitter polariton with exquisite selectivity. Furthermore, we propose that in this regime, the intrinsic losses of the nanocavity-emitter device (which limit the polariton lifetimes) could be precisely compensated for by means of the coherent and non-perturbing, single-mode waveguide feeding at a suitable rate, effectively paving the way towards strongly coupled light-matter states that are robust against decoherence at room temperature. Our scheme contrasts sharply with the conventional belief that preserving an individual quantum state should demand cryogenic cooling in conjunction with strict isolation of the system from the deleterious effects of its environment. Here, we fully embrace dynamic dissipation under ambient conditions, strategically harnessing its interplay with plasmon interference in a specific dressed state to establish the qnCPA regime itself. As a hitherto unexplored paradigm for quantum state preparation and preservation in plasmonic cavity quantum electrodynamics (cQED), qnCPA offers exciting prospects for innovative and room-temperature-viable quantum nanophotonic technologies.

## Results

The use of nanometric metal-insulator-metal gaps, such as that shown in Fig. 1a, constitutes the traditional scheme for achieving near-field strong coupling, where the ultralow mode volume of the plasmonic excitation enhances the cavity-emitter coupling strength beyond the dissipation rates inherent in both subsystems. As a canonical example, we consider a plasmonic nanocavity consisting of a coaxial dimer of gold nanorods with the same diameter, equal lengths $L = 114$ nm and end-to-end separation $d_2 = 3$ nm, whose longitudinal bonding mode exhibits strong subwavelength field localization and enhancement in the gap between them. In order to create a strongly coupled system, as illustrated in Fig. 1a, a quantum emitter (QE) is placed in the center of this gap, and the coupled dimer-emitter system is excited by means of a short optical pulse; the ensuing hybridization of the QE excitation and nanocavity plasmon gives rise to upper and lower polariton states $|u\rangle$ and $|l\rangle$, respectively. We simulate the dynamics of this coupled system using the Maxwell-Bloch method[52] within a finite-difference time-domain (FDTD) framework, a semiclassical approach that has proven particularly successful for active nanoplasmonic devices and cavity-emitter systems, including the spatiotemporal treatment of strong coupling behavior[13,30,53,54]. Details regarding the simulation methodology and numerical parameters relevant to this work can be found in Supplementary Information (SI) (Supplementary Note 6); we briefly mention here that the QE transition is tuned into resonance with

the bonding mode of the nanocavity at 798 nm, and is characterized by a radiative relaxation rate of $\gamma_r = 6.5 \times 10^{-4}$ meV, a pure dephasing rate of $\gamma_d = 26$ meV (half-widths at half-maximum) appropriate to room temperature and a transition dipole moment $\mu = 30$ D oriented along the nanocavity axis. Note that such a high pure dephasing rate at room temperature is essential to achieve strong coupling, since it enhances the spectral overlap between the QE transition and plasmonic mode by matching their linewidths[28,55]. Furthermore, whilst our specific nanocavity-QE system might be regarded as difficult to realize experimentally, its utmost simplicity and the feasibility of coupling a single plasmonic nanowire mode in what follows render it a suitable platform for illustrating the concept of qnCPA. In particular, although the few-nanometer gap size precludes many mesoscopic QEs (such as quantum dots), recently explored near-field transducer designs[31–34] for plasmonic cQED may relax the requirement of such extreme nanocavity-QE proximities, thereby easing practical fabrication constraints.

Figure 2a displays the scattering spectrum of this coupled system under plane-wave irradiation, evidencing two peaks at wavelengths of 773 nm and 823 nm, corresponding to a splitting of 96.5 meV. This is complemented by an analysis of the photoluminescence (PL) lineshape, which is here theoretically modeled via a quantum master equation treatment within the framework of cQED (see SI, Supplementary Note 7). Experimentally, the PL spectrum would be measured by exciting the emitter with a pump laser and collecting the luminescence from the coupled system[31,56,57]. Crucially, the simulated PL spectrum of the isolated nanocavity-QE device in Fig. 2a also presents a double-peak structure, with peak positions in excellent agreement with those in the scattering spectrum. Aside from this peak splitting in the spectral domain, perhaps the most striking manifestation of the strong coupling regime can be found in the temporal dynamics of the QE polarization $P(t)$ (see SI, Supplementary Note 6). Fig. 2b displays $P(t)$ normalized with respect to the incident field amplitude $E_0$. The polarization oscillations exhibit a period of ~ 1 fs, corresponding to the transition frequency of the QE, and show an overall exponentially decaying envelope. Importantly, however, this envelope presents an oscillatory modulation whose characteristic timescale conforms to the observed spectral splitting, and which persists only during the first ~ 120 fs of the nanocavity-emitter interaction. The presence of these Rabi oscillations signifies two rounds of coherent energy cycling between the QE and dimer plasmon mode, and together with the peak splitting in the spectra of Fig. 2a, constitutes an unambiguous signature of the strong coupling regime between the QE and the

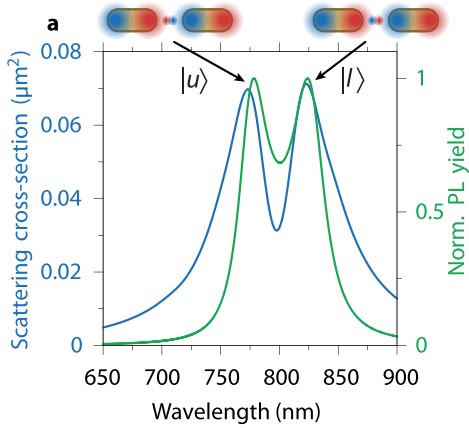
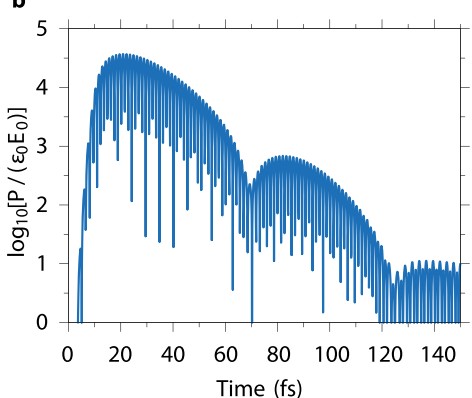
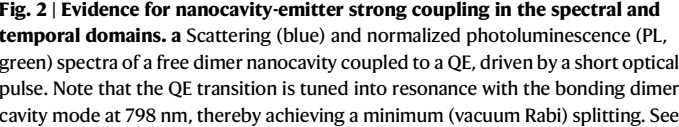

**Fig. 2 | Evidence for nanocavity-emitter strong coupling in the spectral and temporal domains. a** Scattering (blue) and normalized photoluminescence (PL, green) spectra of a free dimer nanocavity coupled to a QE, driven by a short optical pulse. Note that the QE transition is tuned into resonance with the bonding dimer cavity mode at 798 nm, thereby achieving a minimum (vacuum Rabi) splitting. See

the main text for other relevant emitter characteristics. Schematics of the nanocavity-QE system in the upper ($|u\rangle$) and lower ($|l\rangle$) polariton states are shown above the spectra. **b** Temporal dynamics of the normalized polarization $P/(\epsilon_0 E_0)$ of the QE, plotted in a logarithmic representation.

nanocavity plasmon. Nevertheless, the spectral and temporal data in Figs. 2a, b simultaneously reveal two important and hitherto unavoidable limitations. Firstly, such a scheme produces only a superposition of states from the upper and lower polariton branches, with no obvious strategy towards preferentially addressing either of the two light-matter states in isolation upon tuning the QE transition into resonance with the nanocavity mode. Secondly, the strongly coupled states are only short-lived, as reflected by the rapid decay of the Rabi oscillations after only ~ 2 rounds of energy exchange between nanocavity and QE. The lifetimes of the upper and lower polariton states are inevitably limited by the ultrafast decay of the bonding plasmon mode of the dimer, thereby allowing only a correspondingly short temporal window in which the hybrid light-matter states could be exploited for quantum applications.

Here, we propose a unique principle for selectively preparing and protecting strongly coupled light-matter states under ambient conditions, by harnessing the near-field energy transfer and perfect absorption characteristics of gCPA in a lossy plasmonic system (see SI, Supplementary Note 1). We introduce a plasmonic waveguide in the form of a semi-infinite gold nanowire, having the same circular cross-section, diameter (30-nm) and hemispherical end-cap termination as the gold nanorods, coupled to the nanocavity-QE system across a gap of size $d_1$ (see Fig. 1b). In general, the reflection characteristics of the fundamental $TM_0$ mode of the nanowire (arising from its termination) are sensitive to the presence of proximal nanostructures and QEs. Previous theoretical analysis has shown that a gCPA condition can be identified for both the radiant and subradiant plasmonic modes of a single nano-antenna[51], and this treatment can readily be extended to the dimer nanocavity system considered here (see SI, Supplementary Note 4). The gCPA condition is a special one in which the coherent superposition of all waves being reflected back from the nanorod dimer onto the wire exhibits exactly equal amplitude and opposite phase compared to the directly reflected wave at the wire termination, resulting in a perfect destructive interference of back-propagating waves and thus a unidirectional, near-field power coupling from the waveguide into a selected resonance of the nanocavity-QE system. Once the gCPA condition is achieved, the presence of the driving wire becomes cloaked with respect to the nanorod dimer and vice-versa[51]. To achieve gCPA, the resonator coupled to the wire termination must support the correct amount of loss, adjusted here via the length of each antenna rod $L$, while the gap size $d_1$ must be chosen appropriately to transmit a sufficient portion of the mode back into the wire for each round trip[51].

As we demonstrate here however, a powerful feature of gCPA is its generality, being applicable not only to passive nanophotonic devices like the waveguide-coupled nanocavity mentioned above (see SI, Supplementary Note 4) but also to active systems that incorporate quantum matter. In our present context, the introduction of a QE into the nanocavity gives rise to additional, non-radiative, and radiative modal energy loss mechanisms, reflecting its internal excitation and emissive decay characteristics. Nevertheless, the concept of gCPA allows one to focus only on a particular sub-matrix of the global system scattering matrix, namely that which expresses the coupling between the input nanowire and hybrid nanocavity-emitter modes, and to identify a zero-reflection condition for this input mode by zeroing the corresponding eigenvalue[51,58,59] (see SI, Supplementary Note 1). As such, although the presence of a QE certainly impacts the type and relative importance of the different dissipation processes inherent to the system, the major part of the scattering matrix that captures all of the complementary loss channels (particularly the radiative ones) is not explicitly required in the diagonalization. Thus, provided that the impact of these loss channels on the guided mode reflection and transmission coefficients is properly captured (as in our FDTD electrodynamic modeling), both the frequency specificity and effective loss compensation afforded by gCPA can, in principle, resolve the

issues of conventional plasmonic strong coupling schemes highlighted above, allowing plasmon-emitter polaritons to be selectively induced and their coherence protected from antagonistic mechanisms of decay and decoherence under ambient conditions. In this setting, gCPA provides the basis for an unexplored regime that we term qnCPA.

Since the nanowire diameter is sufficiently small, we can restrict our study to the fundamental ($TM_0$) propagating surface plasmon mode with an axially symmetric field profile[60,61]. The mode electric field characteristics and complex propagation constant $k = \beta + i\alpha$ are determined by means of finite-difference frequency-domain calculations performed using commercial software (see SI, Supplementary Note 3), where $\beta = 2\pi/\lambda_{eff}$ is the real propagation constant with $\lambda_{eff}$ the effective wavelength, and $\alpha$ is the field decay constant originating from Ohmic dissipation. This $TM_0$ mode is then employed as a source in three-dimensional Maxwell-Bloch simulations within an FDTD framework, allowing us to explore the coherent driving of the proximal nanocavity-QE system (see SI, Supplementary Note 5).

In order to ascertain the qnCPA condition under waveguide driving, we systematically vary the nanorod lengths, thereby tuning the amount of Ohmic dissipation as well as the relative phases of transmitted and reflected surface plasmon waves across the gaps. Using a combination of semi-analytical transfer matrix and FDTD simulations (see SI, Supplementary Notes 2 and 4), we carefully map the evolution of the nanowire guided-mode reflectance and phase change with the length of the nanorods, and thereby identify a nanocavity geometry that achieves qnCPA for each polariton state. Figure 3a, b display the results of this optimization process, evidencing that at certain critical nanorod lengths, namely 101-nm for $|u\rangle$ and 130-nm for $|l\rangle$, a quasi-Lorentzian lineshape in the guided-mode reflection spectrum emerges with a deep minimum that reaches zero reflectance, accompanied by a phase discontinuity that unambiguously signifies the qnCPA regime. This vanishing of the reflectance under waveguide feeding manifests a unidirectional, near-field energy delivery to the strongly coupled, nanocavity-QE system that selectively excites it in a chosen polariton state. Moreover, the effective cloaking of the waveguide termination via a special, non-bonding condition in qnCPA, inherited from its classical (gCPA) counterpart[51], ensures that the dressed states prepared in this regime are unperturbed by the waveguide driving itself, and should thus possess the characteristics of the pure nanocavity-QE dressed states. It is noteworthy that aside from the polaritonic minima seen in both Fig. 3a, b, another narrow-lineshape feature appears at shorter wavelengths, with an associated steep but nevertheless continuous phase variation. These reflection dips arise from the non-radiative, antibonding mode of the dimer nanocavity, whose field node in the gap region precludes strong coupling with the QE.

In order to elucidate the near-field characteristics of each driven polaritonic state in qnCPA, we perform a frequency-resolved analysis of our finite-pulse simulation data. Specifically, we extract the spatial electromagnetic near-field of the nanocavity-QE system at each polariton excitation frequency in Fig. 3a,b, and present its dynamic evolution in Supplementary Movies 1 and 2 for the upper polariton state, and in Supplementary Movies 3 and 4 for the lower one (see also SI, Supplementary Note 9 for a description of the movies). Our analysis reveals that at the critical frequency for qnCPA, the plasmonic nanowire waveguide and nanocavity-QE system are rendered in a novel kind of impedance matching, where the nanowire excitation is transferred coherently and unidirectionally (i.e., without reflection), inducing a pure, plasmon-emitter dressed state facilitated by the inherent frequency selectivity of qnCPA. Note that driving a nanocavity-QE system via near-field coupling to a plasmonic waveguide contrasts sharply with the standard approach of using ultrashort (and thus very broadband) excitation pulses from far-field optical sources; here, the narrow bandwidth of the qnCPA dips implies that highly broadband excitation is inadequate to properly exploit the qnCPA condition. The two polaritonic states are distinguished by a well-defined phase

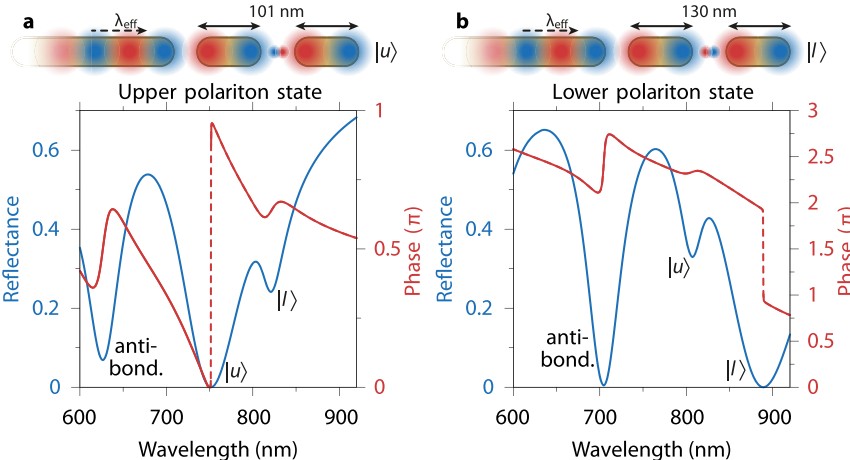

**Fig. 3 | qnCPA in a nanoplasmonic waveguide-driven nanocavity-emitter system. a** Reflectance (blue) and phase change (red) of the $TM_0$ nanowire guided mode in a system optimized for qnCPA with the upper nanocavity-QE polariton state $|u\rangle$ ($L = 101$ nm, $d_1 = 3$ nm and $d_2 = 3$ nm). **b** As in (**a**) but for qnCPA with the lower nanocavity-QE polariton state $|l\rangle$ ($L = 130$ nm, $d_1 = 3$ nm and $d_2 = 3$ nm). The reflectance minimum due to the antibonding nanocavity mode is indicated in each case, and does not correspond to qnCPA. Schematics above illustrate the waveguide-driven nanocavity-QE system for the (**a**) upper and (**b**) lower polaritons.

relationship between the local QE and nanocavity electric fields, as evidenced by the temporal snapshots (magnified in the nanogap-QE region) of the $E_x$ (Fig. 4a, b) and $E_z$ (Fig. 4c, d) components, where the QE near-field is shown enclosed within green circles to distinguish it from the nanocavity near-field. We observe that the local QE and nanocavity fields are in phase for the upper polariton state (Fig. 4a) and out of phase for the lower one (Fig. 4b), as discerned from the matching signs and colors (red corresponding to positive, blue to negative) for the upper polariton and their dissimilarity in the case of the lower polariton.

Importantly, the ability to excite plasmon-emitter polaritons with both exquisite selectivity and in a perturbation-free manner via qnCPA seeds is a potential route to preserving their coherence in time. We conjecture that by subjecting the nanocavity-QE system to continuous-wave driving by a plasmonic waveguide in the qnCPA regime and at a suitable rate, a chosen polaritonic state could be coherently excited and temporally sustained in a condition of exact loss compensation, where the near-field power delivery by the waveguide precisely balances the intrinsic cavity and QE losses. In this respect, qnCPA constitutes more than a merely efficient approach to feeding energy into a polaritonic device; rather, it offers a profound opportunity to render strongly coupled quantum states robust against dynamic dissipation under ambient temperatures, without the need to impose cryogenic conditions or to isolate the quantum system from its physical environment.

## Discussion

In this article, we have proposed a unique principle for selectively initializing and preserving the coherence of strongly coupled light-matter states under ambient conditions, harnessing the coherent, unidirectional, and non-perturbing energy transfer characteristics of near-field qnCPA. By coupling a nanocavity-QE system to a plasmonic nanowire waveguide in the qnCPA regime, we have shown that the upper or lower, plasmon-emitter polaritons can be initialized uniquely, where the lack of a reflection signal specifically at the polariton excitation energy signifies a one-way feeding, locking the strongly coupled system in a chosen dressed state. The intrinsic non-bonding condition in CPA[51] also plays an instrumental role: the effective decoupling of the waveguide termination from the nanocavity-QE system in qnCPA ensures that the polaritons remain unperturbed in the presence of the driving waveguide, and therefore precisely those of the isolated nanocavity-emitter system. Crucially, the very same waveguide feeding

and decoupling mechanisms also unveil a promising route towards preserving the polariton states in time: by continuously feeding the polaritonic system at a suitable rate, the dressed state could be coherently sustained in a dynamic equilibrium, in which the dissipation inherent to the plasmonic resonator and QE are compensated by the non-perturbing, near-field energy delivery. Such a scheme is fundamentally distinct from the simple, resonant driving of a strongly coupled system by means of weak, continuous-wave laser light tuned to a polaritonic transition, which does not achieve stationary dressed-state populations. Indeed, by invoking a rigorous, quantum steady-state analysis (see SI, Supplementary Note 8), we have shown that monochromatic driving of a Jaynes-Cummings system at either of the first-rung polaritonic transition frequencies necessarily induces semi-classical Rabi oscillations, locking the system in an excitation-de-excitation cycle. In contrast, by harnessing our proposed qnCPA scheme, continuous and unidirectional feeding of the nanocavity-QE system via the waveguide could, in principle, facilitate a dynamic compensation of the dissipative mechanisms (losses) inherent to the system, coherently exciting and non-perturbatively-sustaining-a-chosen polariton state without multilevel Rabi oscillations.

In principle, a qnCPA scheme could be operated under the cryogenic conditions typical of today's nascent quantum hardware. Here, the reduced QE linewidths attained at low temperatures (by virtue of suppressed QE dephasing) would necessitate the use of high-$Q$ dielectric cavities, whose photonic modes offer a compatibly narrow linewidth for more efficient light-matter interaction. However, the low-loss character of such resonators itself poses a significant challenge for realizing CPA effects, which rely so crucially on an adequate and often adjustable amount of dissipation. In this respect, plasmonic architectures and ambient-temperature conditions emerge as perhaps the most natural and viable route to achieving qnCPA in practice. Indeed, in contrast to the traditional protocol in cQED of low-temperature operation and strict isolation from the deleterious effects of the environment, our study makes no attempt to mitigate the inevitably severe Ohmic and open-cavity radiation losses of the nanoplasmonic resonator, or the inherent decay and dephasing mechanisms of the emitter. Rather, the intrinsic losses play a fundamental and indispensable role in enabling unidirectional, nanoscale qnCPA. From a methodology perspective, our combination of classical transfer matrix and semiclassical Maxwell-Bloch modeling provides a general, computationally tractable, and physically transparent framework that allows the qnCPA condition in a waveguide-driven nanocavity-QE

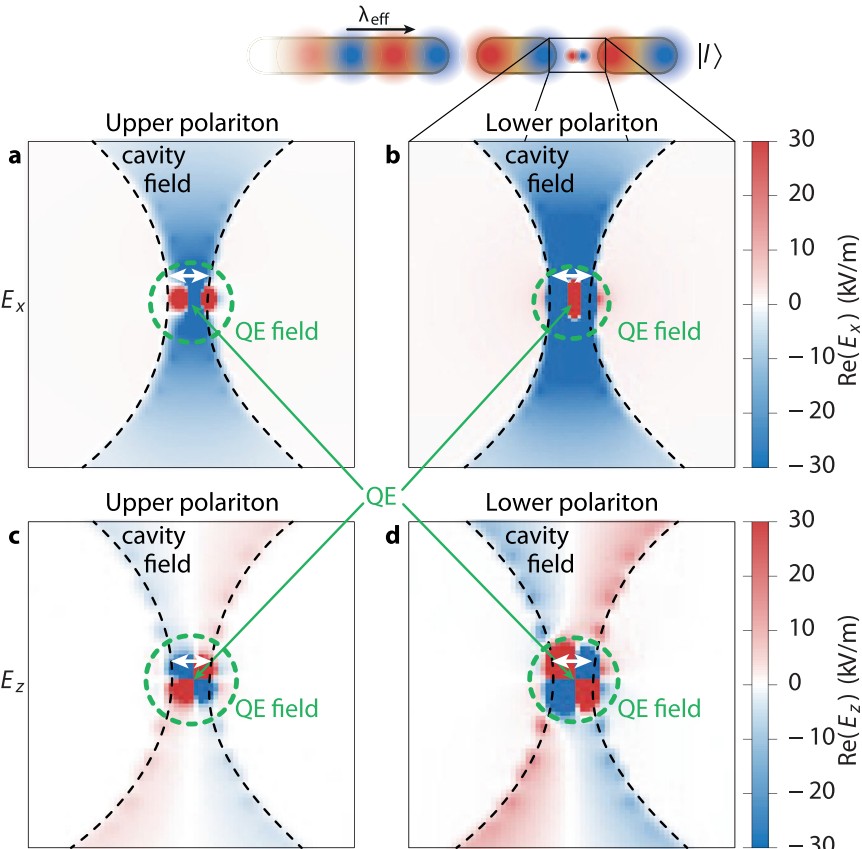

**Fig. 4 | Spatial near-field characteristics of the nanocavity-QE upper and lower polariton states in the qnCPA regime.** Temporal snapshots of the time-harmonic electric near-field components $E_x$ in (**a**, **b**) and $E_z$ in (**c**, **d**), magnified in the nanogap-QE region (marked by black boxes), are displayed at the respective critical wavelengths for qnCPA. The two polariton states are distinguished by the nature of the phase relationship between the local nanocavity and QE fields, namely an in-phase relation for the upper polariton (a,c) and an anti-phase relation for the lower polariton (b,d).

system to be systematically predicted for a desired polariton mode, as well as satisfied to almost any desired degree of accuracy (limited only by the finite spatial and frequency step sizes), merely by simple tuning of the system geometrical parameters.

As a previously unexplored paradigm for initializing quantum light-matter states and preserving their coherence, qnCPA offers compelling opportunities for room-temperature quantum nanophotonic technologies. Perhaps most obviously, the exquisite sensitivity of qnCPA itself renders it a natural candidate for quantum sensing or detection applications; here, the nanocavity-QE system is locked in a chosen dressed state under waveguide driving without reflection, until some external perturbation (such as the absorption of an energetic photon by the strongly coupled system or the presence of a molecular analyte) disturbs the qnCPA condition and gives rise to a finite reflectance signal, which may be measurable by suitable means. Aside from quantum sensing, we also suggest that qnCPA may open a path to as-yet unexplored quantum plasmonic memory schemes, where preserving the coherence of plasmon-emitter entangled states could extend quantum information storage times for quantum computing and networking applications at the nanoscale.

Our proposed scheme is fully compatible with emerging plasmonic nanocircuitry[62–65] and could thus be realized in a chip-scale environment using fully integrated, electrically-operable components. This is especially appealing given that nanowire-guided modes may be selectively excited by purely electrical means[66], which would facilitate on-demand generation of the strongly coupled quantum states of a nanocavity plasmon and QE, whose existence can optionally be rendered transient or long-lived according to whether the system has been engineered for qnCPA. Implementing our qnCPA-enabled

quantum state protection in such integrated and electrically-operable platforms may lead to powerful, on-chip quantum technologies extending from long-lived polariton networks for quantum computing and simulation to quantum optoelectronic devices based on robust, light-induced quantum phases in solid-state systems.

## Data availability
The data that support the findings of this study are available from the Zenodo database[67] and further information can be obtained from the corresponding authors upon request.

## Code availability
Non-commercial codes used in support of this study are available from the corresponding authors upon request.

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

## Acknowledgements

D.D.A.C., A.D., D.W., and O.H. gratefully acknowledge funding from Science Foundation Ireland (SFI) via Grant No. 18/RP/6236. The computational work reported in this article relied on support and infrastructure provided by the Trinity Center for High-Performance Computing, with funding from the European Research Council, Science Foundation Ireland, and the Higher Education Authority through its PRTLI program. P.G., T.He., T.Ho., R.T., and B.H. acknowledge funding by the Deutsche Forschungsgemeinschaft (DFG, German Research Foundation) under Germany's Excellence Strategy through the Würzburg-Dresden Cluster of Excellence on Complexity and Topology in Quantum Matter, ct.qmat (EXC 2147, Project ID ST0462019), as well as through a DFG Reinhard Koselleck project (HE5618/6-1). This work was partly funded within the QuantERA II Program, has received funding from the European Union's Horizon 2020 research and innovation program under Grant Agreement No 101017733, and with DFG (HE5618/12-1), SFI (22/QERA/3821) and NCN (Poland). J.-S.H. acknowledges funding by DFG through SFB-NOA-C1 (398816777) and IRTG-2675-C1 (437527638), and the support from Innovation Project 2020/21 of Leibniz IPHT.

## Author contributions

J.-S.H., B.H., and O.H. conceived and initiated the study. Y.L. performed the FDTD and Maxwell-Bloch simulations under the guidance of D.D.A.C. and O.H. D.D.A.C. and wrote the manuscript with input from all authors. P.G., T.He., T.Ho., and R.T. developed the transfer matrix technique utilized in this study, while P.G. performed the corresponding numerical calculations. A.D. and D.W. prepared the figures. A.D. prepared the visualizations. B.H., and O.H. provided overall supervision of the work.

## Competing interests

The authors declare no competing interests.
