## [Peer Review File · Nature Communications]

Room-temperature Quantum Nanoplasmonic Coherent Perfect AbsorptionREVIEWER COMMENTS

Reviewer #1 (Remarks to the Author):

This article by Y. Lai et al. and entitled "Room Temperature Quantum Coherent Perfect Absorption" is about the study of the qCPA using nanocavities in order to preserve quantum states from a quantum emitter in between the nanoresonators. In order to realise qCPA, an additional 'variable' nanowire is added in order to control the losses to obtain the right interferences for efficient CPA.

This work is only simulations and theoretical models and several results are shown. It is certainly a timely and of great interest work for quantum optics/technologies for preserving quantum coherence.

The work is very-well done with very nice results. However few aspects are missing and several points need to be elucidated before publication and here is the list of issues and questions to be addressed in my view:

1- In the abstract, I do not reckon the use of the word 'immortalization' seems correct or appropriate.

2- the mechanism of CPA when adding the additional nanowire is not crystal clear and perhaps a schematic would be useful.

3- there is a strong issue is the fact that the gap is only 3 nm but quantum dots or any other quantum emitter (part from molecules) will be bigger than this. So this study is nice but unfortunately does not seem very realistic experimentally.

4- coherence is mentioned and is the main part of this study and thus it would be interesting to have a discussion on the consequences on the T1 and T2 of the emitter.

5- again, it is mentioned that a signature of this study would be through the Rabi oscillations but the period is on the order of 70 fs... Again, this seems pretty difficult to observe. Does that mean the scheme works but for less than 100 fs? The fact that it is at room temperature does not help here

6- one major argument is to say that it is working at room temperature but in fact, it seems pretty difficult to observe. Working at low temperature is not a real issue for quantum technologies and as such, would we have better results then?

7- whenever we have blinking (PL or lifetime), is there anything happening on the spectra? Are they modified?

8- it would be nice to have a discussion on the dipole orientation. What would the far-field emission look like?

I would expect these points to be addressed before eventually publishing in Nature Communications at this stage.

Reviewer #2 (Remarks to the Author):

Please see the attached file.

[**Editorial note:** Please see below for the file.]

Dear Authors,

After reviewing the manuscript entitled "Room Temperature Quantum Coherent Perfect Absorption" by Y. Lai et al., submitted for publication in *Nature Communications*, I suggest its major revision. Please consider my comments below.

Key results

The manuscript proposes and theoretically examines the application of coherent perfect absorption (CPA) of classical light to the problem of hybrid light-matter states excitation. Authors consider a strong coupling regime of the emitter and plasmonic nano-cavity, driven through a plasmonic waveguide. The properties of the resulting hybrid light-matter states (polaritons) are studied and compared to the conventional excitation schemes. The key messages and claims in the manuscript are:

- 1) The CPA regime enables the selective excitation of upper and lower hybrid plasmon-emitter states via engineering the plasmonic nano-cavity.
- 2) The CPA regime allows compensation for the dissipation inherent in the plasmonics and, accordingly, allows robust quantum information processing in such systems.
- 3) In the CPA regime, quantum emitters can be operated at room temperature.
- 4) The described scheme can establish an "entirely new approach to optical quantum sensing at room temperature."

Validity

Below, I address these claims point by point.

- 1) The first claim – selective excitation of the plasmon-emitter state, is well-supported by the data and technically sound.
- 2) The second claim is not compelling to me. While CPA allows energy to be fed into the system effectively, locking it in one of the hybrid states, such a system is still of little use in quantum information processing. To be useful, the system should maintain any quantum state, not only one. If one knows the state, there is no need to maintain it. Also, to make some information processing in such a system, one needs to stop feeding light through the CPA mechanism and do something else. In this scenario, the system will experience the same dissipation rate as in any other approach. Finally, the system is not quantum as the plasmon, fed by the classical light, is in the classical state.
- 3) Consequently, the 'room temperature operation' claim is not supported too, as any practical application will require cooling the system down to avoid quick dissipation and perform some useful functions on the system.
- 4) The fourth claim is not supported as the relevant discussion is missing in the text. Also, the phrase "entirely new" is questionable.

Significance

The above critique is mostly concerned with the interpretation of the results, especially in their applicability to quantum information processing, while the results themselves are robust and convincing. These results may be of interest to the community and may contribute significantly to such fields as plasmonics, light-matter interaction, and quantum technologies.

Data and methodology

The methodology and data presentation are convincing and technically sound.

Suggested improvements

I ask the authors to address my comments in the Validity section and comments and questions below.

- 5) The title and, partially, data presentation are not justified. As discussed above, the room temperature operation is not compelling in the context of quantum operation. Also, the

word “quantum” may be misleading as the system under study is composed of *quantum* emitter and *classical* plasmon. I suggest a more specific title.

- 6) I have some concerns about the terminology. CPA assumes coherent (e.g., two-sided) illumination as in the original works by Stone and others. The authors exploit a single-port excitation. Meantime, perfect light absorption under single-side illumination has been known for a long time; it is called a Salisbury screen. In the Salisbury screen, the mechanism of outgoing light cancelation is exactly the same: the first reflected wave interferes destructively with the waves circulating in the structure and leaking out at each round-trip. Could authors comment on this?
- 7) What is the role of the *quantum* nature of the emitter? Would results change if a classical dipole replaces the quantum emitter? At first glance, looking at equations (S2)-(S4) in Supplementary Materials, I expect a similar outcome.
- 8) How does the effect discussed in the manuscript depend on the position of the emitter inside the nano-cavity? Since the efficiency of the emitter excitation depends on its position in the standing wave, this may be an important consideration.

Clarity and context

The structure of the manuscript is well organized, and the ideas are conveyed clearly, except for the parts discussed above. Context-wise, the manuscript is built upon previous works of some of the authors (mostly, Ref. 40 in the manuscript) and is properly placed in the relevant context of plasmonics. Meanwhile, the authors completely ignore studies of quantum effects in CPA, which is clearly relevant to this manuscript. In this regard, I also ask the authors to address the following comment.

- 9) The literature on the quantum effects in coherent perfect absorption is badly reviewed. Firstly, there are papers from the groups of D. Faccio and N. Zheludev, where CPA of different quantum states of light has been studied. A few of them are:
 - a. Roger et al., “Coherent perfect absorption in deeply subwavelength films in the single-photon regime,” *Nat. Commun.* **6**, 7031 (2015)
 - b. Roger et al., “Coherent absorption of NOON states,” *Phys. Rev. Lett.* **117**, 023601 (2016)
 - c. Vetlugin, “Coherent perfect absorption of quantum light,” *Phys. Rev. A* **104**, 013716 (2021)

Also, there are relevant studies on CPA application to excitation and detection in quantum systems:

- d. Akhlaghi et al., “Waveguide integrated superconducting single-photon detectors implemented as near-perfect absorbers of coherent radiation,” *Nat. Commun.* **6**, 8233 (2015)
- e. Everett et al., “Time-reversed and coherently enhanced memory: A single-mode quantum atom-optic memory without a cavity,” *Phys. Rev. A* **98**, 063846 (2018)
- f. Vetlugin et al., “Deterministic generation of entanglement in a quantum network by coherent absorption of a single photon,” *Phys. Rev. A* **106**, 012402 (2022)
- g. Vetlugin et al., “Photon number resolution without optical mode multiplication,” *Nanophotonics* **12**, 505 (2023)

In conclusion, the findings of the manuscript are significant and may be valuable for the community. However, editing and revisiting of the manuscript are necessary.

Kind regards

Response to Reviewer #1

(1) 'In the abstract, I do not reckon the use of the word 'immortalization' seems correct or appropriate.'

We fully appreciate the reviewer's query and are grateful for the opportunity to clarify this core principle proposed in the manuscript. Our assertion that quantum coherent perfect absorption (qCPA) affords an effective "immortalization" of individual, strongly coupled plasmon-emitter states is predicated on the following logical sequence of arguments.

By virtue of its exquisite frequency specificity, qCPA facilitates the selective initialization of a single dressed or polariton state. The lack of a reflection signal specifically at the polariton excitation energy signifies a unidirectional feeding of the strongly coupled nanocavity-emitter system by the nanowire waveguide, locking the former in that chosen polariton state. The intrinsic non-bonding property of CPA, established in our previous work [Grimm *et al.*, *Nanophotonics* **10**, 1879 (2021)], plays an instrumental role here: the effective decoupling of the waveguide termination from the strongly coupled system ensures that the polariton states remain unperturbed in the presence of the driving waveguide, and are therefore precisely those of the isolated nanocavity-emitter system. The very same waveguide feeding and decoupling mechanisms also unveil a pathway towards preserving the dressed state in time. Indeed, we propose that by continuously feeding the polaritonic system via coherent and unidirectional, near-field energy transfer at a suitable rate, it could be rendered in a dynamic equilibrium in which the dissipation inherent to the plasmonic resonator and quantum emitter are exactly compensated by the waveguide energy delivery. The result of this scheme is thus the coherent excitation and non-perturbing preservation of a chosen plasmon-emitter dressed state.

Notably, our work challenges the traditional belief that preserving quantum states and their coherent evolution necessitates cryogenic conditions, together with the strict isolation of the system from the deleterious effects of the environment. Here, we make no attempt to mitigate the inevitably severe Ohmic and open-cavity radiation losses of the nanocavity, or the relaxation mechanisms that impact the quantum emitter dynamics (note that our open-quantum-system modelling incorporates both T_1 and T_2 relaxation processes, see point (4) below). Rather, the intrinsic losses actually play an indispensable role, for they fundamentally enable unidirectional CPA through their interplay with interference.

In response to the reviewer's comment, we have consolidated the above reasoning as a basis for a proposed, qCPA-enabled polariton preservation scheme (see pages 13-16 of the main text). We have also removed the term "immortalization" from the abstract, in order to avoid confusion.

(2) "the mechanism of CPA when adding the additional nanowire is not crystal clear and perhaps a schematic would be useful."

We thank the reviewer for their helpful suggestion to enhance the clarity of our discussion and agree with the benefit of adding an additional schematic figure.

In general, the reflection characteristics of the nanowire guided mode depend sensitively on the terminal conditions of the structure. For a plasmonic dimer nanocavity placed in proximity to the nanowire termination, the condition of generalized coherent perfect absorption (gCPA)

corresponds to a coherent, unidirectional and non-perturbing energy transfer from the nanowire to the dimer in one of its resonant modes. The phenomenon occurs when the directly reflected wave at the nanowire termination bears exactly matching amplitude and opposite phase compared to the coherent superposition of all waves reflected back from the nanorod dimer itself, facilitating a perfect destructive interference of back-propagating signals and an ideal power coupling (impedance matching) between the nanowire and the dimer. In our opinion, the transfer matrix analysis discussed in Supplementary Information (SI), Section S2, offers the simplest and most transparent description of these interference effects, and elucidates the role of both the plasmon propagation-length-dependent phases and dissipation. In particular, by employing this transfer matrix method in conjunction with full-dimensionality finite-difference time-domain (FDTD) simulations, an optimal combination of rod lengths and gap sizes can be precisely determined to achieve the requisite interplay of dissipation and interference for coherent perfect absorption of the incoming plasmon mode. The effect is manifested not only by a zero reflectance, but also by a discontinuity in the phase change upon reflection, and is unambiguously confirmed via the complex wavevector-plane analysis presented in SI, Section S4.

To address the comment by the reviewer, we have extended the discussion of the transfer matrix method in Section S2, and provided a detailed schematic depicting the aforementioned interference effects (see pages 6-8 of the SI). Note that extending these principles to the case of a nanowire-driven polaritonic system (i.e., a strongly coupled dimer-QE system) is a core focus of the manuscript, for which pertinent motivation and numerical evidence can be found in pages 10-12 of the main text.

(3) "there is a strong issue is the fact that the gap is only 3 nm but quantum dots or any other quantum emitter (part from molecules) will be bigger than this. So this study is nice but unfortunately does not seem very realistic experimentally."

We are grateful to the reviewer for highlighting this important issue, and agree that the inter-rod junction considered in our work is prohibitively small for mesoscopic quantum emitters (QEs) like colloidal quantum dots, which are often tens of nanometers in size. It is worth noting that our manuscript features only the generic term "quantum emitter", with the aim of conveying that our proposed qCPA principle is of general significance and is not necessarily limited to a specific nanocavity design or choice of emitter. In particular, whilst a few-nanometer gap size was required to achieve strong coupling conditions for a dimer nanocavity, other recently explored, near-field transducer designs [Groß *et al.*, *Sci. Adv.* **4**, eaar4906 (2018); Bello *et al.*, *Nano Lett.* **20**, 5830 (2020); *Nano Lett.* **22**, 2801 (2022)] may relax the need for such extreme gaps and QE-resonator proximity, thus offering a potential route to realising our proposed qCPA scheme with quantum dots or other suitable excitonic nanomaterials.

In order to acknowledge the above and address the comment of the reviewer, we have added an appropriate statement on page 8 of the main text.

(4) "coherence is mentioned and is the main part of this study and thus it would be interesting to have a discussion on the consequences on the T1 and T2 of the emitter."

The reviewer raises an interesting suggestion here. In the following, we clarify the conditions assumed in our numerical simulations before offering a general comment on the matter.

Our Maxwell-Bloch simulations include both T_1 (longitudinal) and T_2 (transverse) relaxation processes. Specifically, $1/T_1 \sim \gamma_r$, where γ_r is the radiative relaxation rate, while $1/T_2 \sim \gamma_d$

in which γ_d is the total dephasing rate (note that we neglect thermal pumping effects here). Importantly, our study is focused on harnessing qCPA as a means of polariton preparation and preservation at room temperature, where typically $\gamma_d \gg \gamma_r$. Indeed, based on previous experimental and theoretical work concerning room-temperature plasmonic cavity quantum electrodynamics (cQED) in the ensemble- and single-QE regimes [Groß *et al.*, *Sci. Adv.* **4**, eaar4906 (2018); Kongsuwan *et al.*, *Nano Lett.* **19**, 5853 (2019); Yang *et al.*, *Nano Lett.* **24**, 238 (2024)], we adopt a value of $\gamma_d = 4 \times 10^{13}$ rad/s, which is more than four orders of magnitude larger than $\gamma_r = 1 \times 10^9$ rad/s. As such, $T_2 \ll T_1$, and our work thus identifies and explores qCPA in a regime dominated by a T_2 relaxation mechanism. Moreover, it should be underlined that room-temperature conditions in conjunction with inherently lossy plasmonic resonators appear to offer the most natural approach to achieve qCPA, where the broadening facilitated by pure dephasing improves the compatibility of the QE linewidth with the broadband plasmonic nanocavity mode, thereby enhancing the plasmon-QE coupling efficiency for access to the strong coupling regime (see pages 8 and 16 of the main text).

More generally, the disparity between the T_1 and T_2 timescales need not be so stark, and both kinds of relaxation processes can play an important role in influencing the quantum dynamics of a two-level emitter interacting with a nanocavity plasmon mode. A crucial feature of our work, however, is the establishment of a robust protocol for identifying the qCPA state in the presence of a QE with essentially arbitrary T_1 and T_2 characteristics. Indeed, our approach of systematically mapping the reflectance and phase change as a function of geometry, via rigorous mode expansion and Maxwell-Bloch calculations, not only fully incorporates T_1 and T_2 relaxation effects, but remains valid irrespective of their specific timescales or relative importance. The T_1 and T_2 contributions (which are not readily isolated in practice) are manifested only by their impact on the choice of rod lengths and gap sizes, which we adjust in order to ensure the necessary interplay between near-field interference and dissipation for qCPA.

In view of the above, and given that our express purpose is to establish qCPA as a paradigm for quantum nanophotonic applications at room temperature, we have not ventured into a dedicated discussion of T_1 and T_2 relaxation effects, or of specific numerical values for these timescales (except those pertinent to room temperature) in the main text or SI.

(5) "again, it is mentioned that a signature of this study would be through the Rabi oscillations but the period is on the order of 70 fs. Again, this seems pretty difficult to observe. Does that mean the scheme works but for less than 100 fs? The fact that it is at room temperature does not help here."

We agree with the reviewer that such ultrafast Rabi oscillations would be difficult to observe in practice. Indeed, the time-resolved measurement of Rabi signals in plasmonic cQED represents a general challenge, one which demands state-of-the-art femtosecond metrology. Nevertheless, we emphasize that Fig. 2 of the manuscript carries the specific purpose of characterizing strong coupling in the isolated dimer-QE system (i.e., a conventional metal-insulator-metal cavity arrangement with no driving waveguide). Complementary to the spectral data presented in Fig. 2a, the temporal Rabi oscillations in Fig. 2b provide unambiguous evidence for the existence of strong coupling and the emergence of plasmon-emitter dressed states in this elementary scenario, before proceeding to the crux of our work, namely the selective preparation of these dressed states via CPA in a waveguide-driven dimer-QE device and the proposal of a novel principle for prolonging their lifetime via its unique, near-field energy transfer properties. Ultimately, the temporal characteristics of the Rabi oscillations in the isolated dimer-QE system have no bearing on the effectiveness of our

proposed scheme, and are merely investigated to confirm the existence of the target dressed states.

On this basis, we have made no modifications to the main text or SI. Regarding the matter of temperature however, we kindly refer the reader to point (6) below.

(6) "one major argument is to say that it is working at room temperature but in fact, it seems pretty difficult to observe. Working at low temperature is not a real issue for quantum technologies and as such, would we have better results then?"

In principle, our proposed qCPA scheme could be operated under the cryogenic conditions typical of today's nascent quantum hardware. Indeed, for any given environmental temperature (and corresponding QE dephasing rate), the plasmon propagation lengths and phases are systematically adjusted via the geometrical parameters of the system (here, rod lengths and gap sizes), until the characteristic zero-reflectance and phase-discontinuity features are observed, thus signifying the establishment of a qCPA state. However, the narrow QE linewidth attained at low temperatures (by virtue of suppressed QE dephasing) would incur minimal spectral overlap with the broadband plasmonic nanocavity mode, and thereby a reduced coupling efficiency relative to room temperature. As an alternative approach, the high-quality-factor photonic modes of dielectric cavities may offer a compatibly narrow linewidth for more efficient light-matter interaction at low temperatures, but then again, their low-loss character may challenge the realization of CPA effects that rely so heavily on the presence of an adequate, adjustable and often artificially introduced amount of dissipation [Chong *et al.*, Phys. Rev. Lett. **105**, 053901 (2010); Phys. Rev. Lett. **106**, 093902 (2011); Baranov *et al.*, Nat. Rev. Mater. **2**, 17064 (2017)].

More generally however, it should be underlined that our reported study embraces an emergent trend in quantum nanophotonics, where the nanoscale compactness, ultrafast functionality and room-temperature viability of plasmonic nanocavities offer exciting opportunities for innovating photonic quantum hardware and elevating their operation to ambient conditions. Whilst it is now well-established that their ultralow mode volumes facilitate access to the strong coupling regime, the inherently ultrafast decay of plasmonic excitations render the polariton lifetimes correspondingly short, irrespective of the ambient temperature. Indeed, the polariton lifetime is limited by its fastest decaying component (plasmon or QE excitation), so that although the intrinsic QE dephasing is strongly suppressed under cryogenic conditions, the nanoresonator plasmon lifetime is hardly improved and the polariton's existence remains on a femtosecond timescale. In our present work, we show that by harnessing the interplay of surface plasmon interference with the intrinsic system losses (which are inevitably severe at room temperature), qCPA emerges as a hitherto unexplored mechanism for preparing hybrid plasmon-emitter states and in principle, rendering them robust against dissipation.

To address this comment by the reviewer, we have added appropriate text concerning the feasibility of low-temperature operation on page 16 of the main text.

(7) "whenever we have blinking (PL or lifetime), is there anything happening on the spectra? Are they modified?"

We are grateful to the reviewer for raising this issue and agree that photoluminescence (PL) intermittancy represents an important issue in practice for plasmonic cQED studies featuring molecular and solid-state QEs. The impact of PL blinking on the spectral manifestations of single-QE strong coupling has been discussed previously in connection with quantum dots [for

example, Radulaski *et al.*, Adv. At. Mol. Opt. Phys. **66**, 111 (2017)], and in principle, our present cQED model for PL could be extended in a similar spirit. However, it is worth noting that in a number of experimental works, including single-QE and ensemble strong coupling studies reported by a number of us [Chikkaraddy *et al.*, Nature **535**, 127 (2016); Groß *et al.*, Sci. Adv. **4**, eaar4906 (2018); Yang *et al.*, Nano Lett. **24**, 238 (2024)], blinking has not been found detrimental to the spectroscopic identification of polaritons (this being the sole purpose of Fig. 2b in the manuscript). Moreover, it is interesting to note that coupling single QEs to plasmonic nanocavities can achieve non-blinking PL; in particular, using a nanocavity formed by a dimer of gold nanorods (similar to our own work), Wang *et al.* [Nano Lett. **24**, 1761 (2024)] have very recently shown that Purcell-enhanced radiative decay can outcompete non-radiative Auger processes that may otherwise promote intermittent PL suppression. As such, the plasmonic platform may offer inherent benefits in mitigating blinking behaviour, complementary to more traditional approaches relying on, for instance, suitable core-shell engineering of the quantum dot.

On this basis, we have made no modifications to the main text or SI.

(8) "it would be nice to have a discussion on the dipole orientation. What would the far-field emission look like?"

In our simulations, we have chosen an orientation of the QE transition dipole moment μ compatible with the intracavity field direction (i.e., longitudinal, along the dimer axis). This ensures an optimal plasmon-emitter coupling strength and the formation of well-defined polariton states, which is an obvious prerequisite for our qCPA-enabled initialization and proposed coherence preservation strategies. Of course, for any perpendicular orientation of μ , the coupling strength is minimised and the system exists merely in the weak coupling regime. This is evidenced in Fig. 1, which compares the scattering spectra for two such QE dipole orientations. Clearly, for perpendicular orientation, no peak splitting effect is observed and the scattering lineshape matches that of the empty dimer cavity. Our proposed qCPA scheme is of little relevance in this case, as no polaritons are formed for subsequent excitation and temporal prolonging via qCPA.

Figure 1: Comparison of scattering spectra for the coupled dimer-emitter system for dipole orientations parallel ($\mu_{||}$) and perpendicular (μ_{\perp}) to the dimer axis. In the case of a perpendicular orientation, there is no observable peak splitting and the scattering lineshape matches that of the empty dimer cavity.

For arbitrary orientations between these two extremes, we underline that the general principles of our analysis remain pertinent. For a fixed orientation of μ , the nanorod lengths and gap sizes can be systematically varied until the emergence of zero-reflectance and phase-discontinuity behaviour is identified, signifying the qCPA regime. However, the obvious need for well-defined and spectrally resolved polaritons naturally favours the parallel alignment treated in our manuscript.

The reviewer also raises the interesting suggestion of examining the far-field radiation patterns. Based on the established physical characteristics of the qCPA regime (see also point (1) above), we predict that the effective decoupling of the nanowire termination from the polaritonic dimer-QE device, in conjunction with the negligible radiation yield from the nanowire termination itself, will ensure that the far-field radiation properties of the polaritonic system match those of its isolated counterpart (these are, in turn, largely inherited from the bonding mode of the dimer). Numerical confirmation of this behaviour would demand the use of very long driving pulses from the waveguide in our full-dimensionality, self-consistent Maxwell-Bloch simulations together with a precise, frequency-domain sampling and propagation of the field distributions at the polariton excitation wavelengths; in practice, the problem has remained computationally intractable with our current computing resources but continues to be a focus of ongoing work.

On the basis of the above remarks, we have made no modifications to the main text or SI.

Response to Reviewer #2

(1) "The first claim - selective excitation of the plasmon-emitter state, is well-supported by the data and technically sound."

We are grateful to the reviewer for their positive appraisal of this claim.

(2) "The second claim is not compelling to me. While CPA allows energy to be fed into the system effectively, locking it in one of the hybrid states, such a system is still of little use in quantum information processing. To be useful, the system should maintain any quantum state, not only one. If one knows the state, there is no need to maintain it. Also, to make some information processing in such a system, one needs to stop feeding light through the CPA mechanism and do something else. In this scenario, the system will experience the same dissipation rate as in any other approach. Finally, the system is not quantum as the plasmon, fed by the classical light, is in the classical state."

We appreciate the reviewer's point and wish to clarify the prospective value of our proposed qCPA scheme for practical quantum technologies. Perhaps most obviously, the intrinsic sensitivity of qCPA itself renders it a natural candidate for quantum sensing or detection applications. Here, the nanocavity-QE system is locked in a chosen dressed state under waveguide driving without reflection, until some external perturbation (such as the absorption of an energetic photon by the strongly coupled system or the presence of a molecular analyte) disturbs the qCPA condition and gives rise to a finite reflectance signal, which could be measured by an appropriate means. Aside from quantum sensing, we also suggest that qCPA may open a path to as-yet unexplored quantum plasmonic memory schemes, where preserving the coherence of plasmon-emitter entangled states could extend quantum information storage times for quantum computing and networking applications at the nanoscale.

Accordingly, we have added appropriate discussion of these potential applications to the main text (see pages 16-17).

The reviewer makes a valid observation regarding the semiclassical nature of our modelling approach, a feature which is indeed made explicit in both the main text and SI. Our choice of the semiclassical Maxwell-Bloch methodology has been informed by a number of important considerations. We begin by noting that a rigorous, quantum theory of the electromagnetic field in the presence of dissipative and dispersive media is well-established in the form of macroscopic quantum electrodynamics [Gruner and Welsch, *Phys. Rev. A* **53**, 1818 (1996); Dung *et al.*, *Phys. Rev. A* **57**, 3931 (1998)]. However, most applications to date have entailed a perturbative treatment of the light-matter interaction, in which the photonic degrees of freedom are removed from the Heisenberg equations of motion describing the two-level operator dynamics. For applications beyond the perturbative regime, such as the strongly coupled plasmon-emitter system discussed in our present work, the electromagnetic degrees of freedom cannot be removed from the quantum dynamical equations in this manner, and the numerical solution of the equations of motion in their full complexity is cumbersome. Recently, a second quantization scheme for the quasinormal modes of dissipative resonators has been proposed [Franke *et al.*, *Phys. Rev. Lett.* **122**, 213901 (2019)] and applied to a number of problems in plasmonic cQED. However, the validity of the quasinormal mode regularization scheme adopted therein has been questioned [Sauvan *et al.*, *Opt. Exp.* **30**, 6846 (2022)], and a dedicated analysis of extended systems (like the plasmonic waveguide considered in our study) has yet to be reported within this framework. Whilst the lack of a

widespread consensus on a truly rigorous quantum theory of quasi-normal modes represents an important and pressing contemporary issue, it falls beyond the rational scope of our present work. Finally, and beyond difficulties at the level of formalism, it is wholly anticipated that time-domain quantum simulations will impose a computational cost that scales exponentially with the number of plasmonic modes and QE states, which represents a particular challenge for plasmonic systems that often present numerous, spectrally overlapping resonances.

Recognizing these issues, we believe that the semiclassical Maxwell-Bloch approach addresses the need for both an insightful and robust computational modelling approach that incorporates the key physical attributes of qCPA. By solving Maxwell's equations self-consistently with the optical Bloch equations for a two-level QE, our study incorporates the electrodynamic complexity of the nanoplasmonic environment (via FDTD calculations) as well as both the coherent and incoherent facets of the QE dynamics (via the open-quantum system treatment). The semiclassical analysis also complements well our previous report on generalized coherent perfect absorption and its characterization in terms of classical field scattering and reflection signatures, while a fully quantum description may ultimately be formulated in terms of intensity correlation functions, and as such, may appear less transparent to the general reader in this first, exploratory study. Lastly, we note that our Maxwell-Bloch approach has been shown in several previous works to provide key insight into the spatiotemporal characteristics of plasmon-emitter strong coupling [Kongsuwan *et al.*, *Nano Lett.* **19**, 5853 (2019); Xiong *et al.*, *Adv. Opt. Mater.* **10**, 2200557 (2022)], and has yielded predictions in favourable agreement with experimental studies featuring plasmonic nanocavities [Kongsuwan *et al.*, *ACS Photonics* **5**, 186 (2018); Yang *et al.*, *Nano Lett.* **24**, 238 (2024)].

We underline that the semiclassical nature of our simulation methodology is declared explicitly in the main text, together with appropriate references (as cited above) that demonstrate its predictive value in the context of plasmonic cQED and quantum device applications (see page 8). The formalism has also been adequately detailed in the SI (see pages 14-16).

(3) "Consequently, the 'room temperature operation' claim is not supported too, as any practical application will require cooling the system down to avoid quick dissipation and perform some useful functions on the system."

We kindly refer the reader to our response to point (6) of Reviewer #1 above, where the matter of room-temperature operation was explicitly raised and any relevant modifications to the main text or SI discussed.

(4) "The fourth claim is not supported as the relevant discussion is missing in the text. Also, the phrase "entirely new" is questionable."

We agree with the reviewer that the claim of optical quantum sensing as a potential application is not directly substantiated by the simulation data and analysis presented in the manuscript. However, it does arise as a very plausible one, and a classical sensing scheme based on gCPA in plasmonic nanoantennas has already been explored in our previous work [Grimm *et al.*, *Nanophotonics* **10**, 1879 (2021)]. Our intention here was to underline that the rather pronounced sensitivity of the qCPA condition to environmental disturbances (particularly to changes in the terminal conditions of the waveguide) represents a particularly interesting route to optical quantum sensing at the single-photon level. As mentioned in connection with point (2) for instance, the absorption of a single, energetic photon from the

ambient environment could excite the strongly-coupled system (initialized in some chosen dressed state) to a higher rung of the Jaynes-Cummings ladder, thereby disrupting the qCPA regime (i.e., shifting the reflectance zero in the complex wavevector plane away from the guided-mode dispersion curve) and producing a non-zero reflectance, which is potentially measurable by an appropriate means.

To avoid any potential misunderstanding, we have removed the explicit reference to quantum sensing (including the phrase "entirely new") from the abstract and added a clarifying statement in discussing the implications of our work (see pages 16 – 17 of the main text).

(5) "The title and, partially, data presentation are not justified. As discussed above, the room temperature operation is not compelling in the context of quantum operation. Also, the word "quantum" may be misleading as the system under study is composed of quantum emitter and classical plasmon. I suggest a more specific title."

We kindly refer the reader to points (2) and (3) above, in connection with the use of the term "quantum" and the claim of room-temperature operation respectively, together with any relevant modifications to the main text and SI.

(6) "I have some concerns about the terminology. CPA assumes coherent (e.g., two-sided) illumination as in the original works by Stone and others. The authors exploit a single-port excitation. Meantime, perfect light absorption under single-side illumination has been known for a long time; it is called a Salisbury screen. In the Salisbury screen, the mechanism of outgoing light cancellation is exactly the same: the first reflected wave interferes destructively with the waves circulating in the structure and leaking out at each round-trip. Could authors comment on this?"

The reviewer raises an interesting comment here. Firstly, we emphasize that coherence plays a role that is just as instrumental to a one-port scheme as it is in the two-port schemes conventionally discussed in the CPA literature, and that there is no fundamental requirement that a CPA scheme should be multiport (as an example, see the work of Jin and Yu [Opt. Exp. **28**, 35108 (2020)]). In the case of our waveguide-driven dimer-QE system (which might be loosely regarded as a one-port scheme, see below), the exact destructive interference between the directly reflected wave at the waveguide termination, and the superposition of all waves reflected back from the dimer-QE device, is facilitated by their coherent character and the existence of a well-defined phase relationship (specifically, an out-of-phase one) between them, ensuring a complete suppression of back-reflection signals.

We note that some caution is warranted regarding the comparison of our present scheme with the dielectric Fabry-Perot-like resonators discussed in the traditional CPA literature. In the latter case, the multiple reflections which accompany the Fabry-Perot resonances can lead to perfect destructive interference between the first reflected wave and all subsequent outgoing waves, with the primary dissipation mechanism being heating (i.e., non-radiative). The CPA effect in such scenarios can be fruitfully analyzed in terms of a finite number of input and output modes, and formally corresponds to a zero eigenvalue of the system scattering matrix and associated phase singularities in the complex wavevector plane. In contrast however, plasmonic resonators exhibit both radiative and non-radiative loss channels, so that the absorbed energy can be lost to the far-field or converted to heat, where the former possibility is largely neglected in the original concept of CPA. The formalism of gCPA adopted in our work substantiates the notion of perfect absorption in scenarios where analysis of a few input and output guided modes is inadequate for a proper understanding; indeed, for plasmonic resonators, it allows one to describe and quantify the selective absorption of a

given input mode in the presence of (infinitely many) radiative loss channels, without the need to fulfill the requirements of CPA for the complete system. In particular, we identify a zero-reflectance condition for the selected input mode by evaluating and zeroing an eigenvalue of the corresponding submatrix of the complete scattering matrix, treating lossy re-radiation via complementary output channels. Although our approach follows the logic of Stone's CPA, it is much more akin to the generalized theory of reflectionless scattering modes that was more recently proposed by Sweeney, Stone and coworkers [Sweeney *et al.*, Phys. Rev. A **102**, 063511 (2020); Stone *et al.*, Nanophotonics **10**, 343 (2021)].

Ultimately, we agree with the reviewer that highly efficient absorption can be realized via multilayer devices, like the Salisbury screen and anti-reflection coatings, that generate destructive interference by means of suitably chosen optical path lengths in the constituent media. However, (g)CPA constitutes a more profound phenomenon, intimately linked to the spectral properties (in a mathematical sense) of the scattering matrix and which cannot be proven merely through an apparently suppressed back-reflection alone. Indeed, we have been especially meticulous in establishing unambiguous evidence for gCPA in the plasmonic cQED context, viz. a complex wavevector plane analysis for the waveguide-driven nanocavity (see SI, Section S4) and the identification of a finite discontinuity in the argument of the reflection coefficient for the qCPA states (see Fig. 3 of the main text).

In response to this comment, we have made no modifications to the main text or SI. In particular, the salient features of gCPA for plasmonic systems and its connection to the formalism of Sweeney, Stone and coworkers was already discussed in the SI (see Section S1 therein).

(7) "What is the role of the quantum nature of the emitter? Would results change if a classical dipole replaces the quantum emitter? At first glance, looking at equations (S2)-(S4) in Supplementary Materials, I expect a similar outcome."

The quantum nature of the emitter is of fundamental significance here. Our present manuscript is intended to present a novel principle for preparing and prolonging the lifetime of strongly coupled, quantum light-matter states, whose formation can only be legitimately discussed in the context of a QE excitation that hybridizes with the nanocavity plasmon. We do agree that the gCPA analysis of Ref. [40] in the original manuscript could be adapted to the case of a classical dipolar emitter coupled to a nanocavity. However, this would correspond to a purely classical scenario, and would not be satisfactory towards meeting our core purpose stated above.

On this basis, we have made no modifications to the main text or SI.

(8) "How does the effect discussed in the manuscript depend on the position of the emitter inside the nano-cavity? Since the efficiency of the emitter excitation depends on its position in the standing wave, this may be an important consideration".

We agree that, in general, the precise spatial location of the QE in the nanogap has an important bearing on achieving CPA. In the specific context of our work, the QE location is constrained to be an intracavity field hotspot (i.e., the center of the dimer gap) such as to ensure a maximal coupling strength and the formation of well-defined plasmon-emitter polaritons, the latter being an obvious prerequisite for our scheme. Small deviations in the QE position from this optimal one will impact the polaritonic splitting and thus the condition for attaining qCPA, but this does not represent a fundamental problem.

On this basis, we have made no modifications to the main text or SI.

(9) "The literature on the quantum effects in coherent perfect absorption is badly reviewed. Firstly, there are papers from the groups of D. Faccio and N. Zheludev, where CPA of different quantum states of light has been studied. A few of them are...Also, there are relevant studies on CPA application to excitation and detection in quantum systems..."

We are very grateful to the reviewer for the opportunity to improve our manuscript in this respect.

We have gladly extended the introduction, adding further contextualization in terms of recent efforts to elucidate and harness coherent perfect absorption in the quantum domain (see page 5). In so doing, we have cited all of the works kindly suggested by the reviewer, as well as several others of pertinence to the discussion.

REVIEWERS' COMMENTS

Reviewer #2 (Remarks to the Author):

Please see the attached file.

[**Editorial note:** Please see below for the file.]

Dear Authors,

Thank you for your thoughtful consideration of the comments. While I believe that the revised manuscript is well aligned with the standards of *Nature Communications*, some revisions are still required.

I acknowledge the responses to my concerns, with the exception of points (2) and (5). Please find my comments below.

- The argument in point (2) is not convincing. While I agree with the utility of the scheme for sensing, I struggle with its applicability for “as-yet unexplored quantum plasmonic memory schemes, where preserving the coherence of plasmon-emitter entangled states could extend quantum information storage times for quantum computing and networking applications at the nanoscale.” Quantum memory is based upon the interaction of quantum light with quantum matter, wherein a strong classical drive is not compatible. Therefore, plasmon-emitter entangled states appear to be beyond the scope of this study.

Furthermore, I believe there is a misunderstanding regarding one of my comments – “Finally, the system is not quantum as the plasmon, fed by the classical light, is in the classical state.” I do not question the validity of the Maxwell-Bloch approach in solving the problem, but I sought to clarify that due to the classical driving field (coherent state of the laser), the entire system operates in the semi-classical, not quantum regime.

- In response to point (5), I must express strong disagreement with the chosen title. The phrase “room temperature coherent perfect absorption” appears illogical to me, as “coherent perfect absorption” is a mechanism or protocol of light absorption that does not have such characteristics as temperature. Compare it, for instance, with “light absorption at room temperature,” “light diffraction at room temperature,” or “light propagation at room temperature.”

Additionally, CPA is not an objective by itself but rather a tool to selectively excite and maintain the system in a particular state. Given that the paper seeks to demonstrate the feasibility of room-temperature operation of the nanoplasmonic quantum device facilitated by CPA excitation, a more appropriate title might be “Room temperature quantum nanoplasmonics enabled by CPA” or a similar variation.

Also, please double-check the “Video—Ex at lower polarization state” file—the color intensities are slightly off.

Kind regards

Response to Reviewer #2

(1) "The argument in point (2) is not convincing. While I agree with the utility of the scheme for sensing, I struggle with its applicability for "as-yet unexplored quantum plasmonic memory schemes, where preserving the coherence of plasmon-emitter entangled states could extend quantum information storage times for quantum computing and networking applications at the nanoscale". Quantum memory is based upon the interaction of quantum light with quantum matter, wherein a strong classical drive is not compatible. Therefore, plasmon-emitter entangled states appear to be beyond the scope of this study."

We are grateful to the reviewer for the opportunity to clarify this important point. Whilst it is indeed the case that our simulation methodology is semiclassical, and that the coherent driving fields from the nanowire waveguide are thus treated in a classical manner, our express intention is to demonstrate the potential of nanoplasmonic coherent perfect absorption (CPA) as an enabler of quantum (rather than semiclassical) technologies.

Given the present lack of a truly rigorous, widely accepted and fully quantized treatment of active nanoplasmonic systems (as discussed in our previous response letter), we have adopted the semiclassical MaxwellBloch approach within a finite-difference time-domain framework, which offers the crucial benefits of capturing the full electrodynamic complexity of the problem as well as computational tractability. However, our use of a semiclassical approach featuring purely classical driving fields should not be taken to imply that our investigated CPA scheme is intended for semiclassical device functionalities. On the contrary, our numerical simulation approach provides a very well-established, physically transparent and computationally manageable means by which to elucidate a previously unexplored paradigm for initializing and prolonging the lifetime of strongly coupled states in plasmonic cavity quantum electrodynamics, whose formation and spatiotemporal character can fruitfully be studied in this manner. Furthermore, given our view towards the novel technologies that quantum nanoplasmonic CPA might facilitate, we believe it prudent to suggest potential quantum information processing applications that would hinge on the possibility of generating, on demand, plasmon-emitter entangled states, such as in quantum sensing and nascent quantum plasmonic memory schemes. Inevitably, investigation of quantum nanoplasmonic CPA at the few- or even single-plasmon level, including characterization of the non-classical plasmon statistics, demands transcending the semiclassical approximation. This remains a matter of active research and will constitute the subject of a future publication.

In response to this comment, we have made no modifications to the main text or supplementary material.

(2) "In response to point (5), I must express strong disagreement with the chosen title. The phrase "room temperature coherent perfect absorption" appears illogical to me, as "coherent perfect absorption" is a mechanism or protocol of light absorption that does not have such characteristics as temperature. Compare it, for instance, with "light absorption at room temperature", "light diffraction at room temperature", or "light propagation at room temperature".

Additionally, CPA is not an objective by itself but rather a tool to selectively excite and maintain the system in a particular state. Given that the paper seeks to demonstrate the feasibility of room-temperature operation of the nanoplasmonic quantum device facilitated by CPA excitation, a more appropriate title might be "Room temperature quantum nanoplasmonics enabled by CPA" or a similar variation."

We are grateful to the reviewer for underlining this issue. Our inclusion of the phrase "room temperature" in the title reflects the ambient-temperature viability of our proposed, quantum nanoplasmonic CPA scheme, where the nanocavity-emitter dressed states are initialized and potentially sustainable in dynamic equilibrium without cryogenic cooling or strict environmental isolation. However, we fully agree with the remark by the reviewer that CPA is "a mechanism or protocol of light absorption that does not have such characteristics as temperature".

Furthermore, we accept the reviewer's suggestion to modify the title, henceforth adopting "Room-temperature Quantum Nanoplasmonic Coherent Perfect Absorption". We believe that this choice mitigates potential ambiguity concerning classical or quantum electrodynamic effects in CPA, while continuing to emphasize the room-temperature operation and nanoplasmonic character of the system, as well as being comparably compact with the original title. Correspondingly, we now also use the abbreviation "qnCPA" for this phrase, as opposed to "qCPA" in the original manuscript.

In response to this comment, we have modified the title of the manuscript to read "Room-temperature Quantum Nanoplasmonic Coherent Perfect Absorption", and have also introduced the corresponding abbreviation "qnCPA" for this phrase throughout.

(3) "Also, please double-check the "Video-Ex at lower polarization state" file - the color intensities are slightly off."

We are grateful to the reviewer for highlighting this issue and have corrected the visualization accordingly.